# FEDERATED LEARNING EMPOWERED BY GENERATIVE CONTENT

## ABSTRACT

Federated learning (FL) enables leveraging distributed private data for model training in a privacy-preserving way. However, data heterogeneity significantly limits the performance of current FL methods. In this paper, we propose a novel FL framework termed FedGC, designed to mitigate data heterogeneity issues by diversifying private data with generative content. FedGC is a simple-to-implement framework as it only introduces a one-shot step of data generation. In data generation, we summarize three crucial and worth-exploring aspects (budget allocation, prompt design, and generation guidance) and propose three solution candidates for each aspect. Specifically, to achieve a better trade-off between data diversity and fidelity for generation guidance, we propose to generate data based on the guidance of prompts and real data simultaneously. The generated data is then merged with private data to facilitate local model training. Such generative data increases the diversity of private data to prevent each client from fitting the potentially biased private data, alleviating the issue of data heterogeneity. We conduct a systematic empirical study on FedGC, covering diverse baselines, datasets, scenarios, and modalities. Interesting findings include (1) FedGC consistently and significantly enhances the performance of FL methods, even when notable disparities exist between generative and private data; (2) FedGC achieves both better performance and privacy-preservation. We wish this work can inspire future works to further explore the potential of enhancing FL with generative content.

## 1 INTRODUCTION

Federated learning (FL) is a privacy-preserving machine learning paradigm that enables multiple clients to collaboratively train a global model without directly sharing their raw data (McMahan et al., 2017; Kairouz et al., 2021). With the increasing concerns about privacy, FL has attracted significant attention and has been applied to diverse real-world fields such as natural language processing, healthcare, finance, Internet of Things (IoT), and autonomous vehicles (Yang et al., 2019).

Data heterogeneity presents a prominent and fundamental challenge in FL, significantly impacting FL's overall performance (McMahan et al., 2017; Hsu et al., 2019). This heterogeneity arises inherently due to the varied environments and preferences in which clients' datasets are collected. Consequently, it results in biased and divergent local models, posing difficulties in achieving a well-generalized aggregated model capable of effectively addressing diverse data sources.

Addressing this issue, many optimization-based works are proposed from diverse perspectives (Wang et al., 2021). On the client side, they regularize the distance between local and global model (Li et al., 2020b; Acar et al., 2020), introduce control variates to correct local gradients (Karimireddy et al., 2020), align the feature space (Ye et al., 2022; Li et al., 2021). On the server side, they introduce momentum to update global model (Reddi et al., 2020; Hsu et al., 2019), adjust the process of aggregating local models (Wang et al., 2020b; Jhunjhunwala et al., 2022), modify model initialization (Nguyen et al., 2022; Chen et al., 2022). However, the performance of all these methods is still severely limited as data heterogeneity fundamentally exists.

In this paper, we propose a new idea of fundamentally mitigating the effects of data heterogeneity with the help of diverse generative content. To realize this idea, we propose a novel framework, Federated Learning with Generative Content (FedGC). In FedGC, each client uses a publicly available generative model conditioned on task-related prompts to generate diverse data, which supplements

the originally client-specific (the root of data heterogeneity) data. The supplemented dataset can subsequently facilitate client model training by encouraging the local model to also learn diverse patterns rather than only patterns of its private data. Despite the simplicity, FedGC can significantly mitigate data heterogeneity as generative diverse data introduces informative and general patterns, thus preventing each client from over-fitting its potentially biased private data.

Furthermore, FedGC is a flexible framework with multiple potential directions. Considering generation efficiency, data diversity, and data fidelity, we summarize four critical aspects in FedGC, including budget allocation, prompt design, generation guidance, and training strategy. In each aspect, we propose three representative solutions as candidates. For example, to achieve a better trade-off between diversity and fidelity during generation, we propose real-data-guidance which generates data conditioned on real data and task-related prompts simultaneously.

To prove the effectiveness of FedGC and deepen understanding, we conduct a systematic empirical study from diverse perspectives, including compatibility with FL baselines, different datasets, different modalities, and different data heterogeneity types; and have several interesting findings. 1) Adding generative data is a more direct, concise, and effective solution to tackle data heterogeneity, than many sophisticated algorithm designs. 2) FedGC can achieve both better privacy preservation and performance. 3) Despite failing to resemble real data, generative data still contributes to enhanced performance as it can implicitly reduce data heterogeneity and model divergence.

Our contributions are as follows:

1. We propose FedGC, a new, simple yet effective FL framework that handles data heterogeneity from a new perspective: generating diverse data to supplement private real data.

2. We summarize four critical and worth-exploring facets in FedGC and propose three solution candidates for each, underscoring its flexibility and potential for future explorations.

3. We provide a systematic empirical study on FedGC framework, showing its effectiveness for tackling data heterogeneity and providing new insights for future works through several interesting experimental findings.

## 2 RELATED WORK

**Federated learning** (FL) enables multiple clients to collaboratively train a global model without sharing raw data (McMahan et al., 2017), which has attracted much attention due to its privacy-preserving property (Li et al., 2020a; Kairouz et al., 2021). Data heterogeneity is one representative challenge in FL that significantly limits the FL's performance (Hsu et al., 2019; Li et al., 2019). Addressing this, many methods are proposed to mitigate its adverse effects from the perspective of optimization. (1) On client-side optimization, FedProx (Li et al., 2020b) and SCAFFOLD (Karimireddy et al., 2020) propose to conduct model-level correction such as regularizing $\ell_2$ distance between local and global model and introducing a control variate to correct gradient of local model. MOON (Li et al., 2021) and FedDecorr (Shi et al., 2022) propose to regularize feature space. (2) On server-side optimization, FedNova (Wang et al., 2020b) and FedDisco (Ye et al., 2023) propose to modify aggregation weights to obtain better-aggregated model. (Nguyen et al., 2022; Chen et al., 2022) explore the effects of model initialization. FedAvgM (Hsu et al., 2019) and FedOPT (Reddi et al., 2020) apply momentum-based optimization to improve global model updating.

Unlike these optimization-level methods that still fundamentally suffer from data heterogeneity, our FedGC framework focuses on data-level improvement, which mitigates heterogeneity of the distributed real data by complementing it with diverse generative data. Besides, our FedGC framework is orthogonal to these methods, allowing seamless integration within our framework.

**Generative models** have demonstrated remarkable performance across multiple domains such as large language models (Ouyang et al., 2022; OpenAI, 2023; Touvron et al., 2023) for language generation and diffusion models (Nichol et al., 2022; Rombach et al., 2022; Saharia et al., 2022) for image generation. Though these models can generate high-quality data for general cases, the generated data is not sufficient to train a well-perform model due to its incapability of representing real data (He et al., 2022), especially for uncommon cases such as medical tasks (Eysenbach et al., 2023; Celard et al., 2023). Recently, (Shipard et al., 2023) shows the importance of data diversity for zero-shot image classification tasks.

In this paper, we systematically explore the potential of using generative models to assist FL on private data. Based on our FedGC framework, we verify that despite failing to fully represent real data, generated data can still contribute to improving the performance of FL under heterogeneous private data. Besides, FedGC is applicable to both image and text regimes.

## 3 Federated Learning with Generative Content

We propose FedGC, a new FL framework that leverages generative content to tackle the issue of data heterogeneity in FL. Based on FedGC, we summarize four aspects worth exploring and propose three methods for each aspect, which serves to provide more insights for future works.

### 3.1 FedGC Framework Overview

Our FedGC follows the standard FedAvg framework, encompassing of four iterative phases: global model broadcasting, local model training, local model uploading, and global model aggregation. Our goal is to generate diverse data to supplement private data to facilitate local model training. Considering communication cost and flexibility, we generate data on the client (local) side, which avoids additional communication cost required for server-to-client transmitting generative data, and enables using the local data as prior to generate more specific data. Thus, we focus on local model training, which is decomposed into: data generation and local model training. Specifically, in FedGC, we 1) design to generate diverse data, 2) merge the generative and private dataset, and 3) train the local model, where the first two are required for only once; see Figure 1 for the overview. Although FedGC is versatile across modalities, our illustration herein will on images for easier understanding.

### 3.2 Data Generation in FedGC

On the designs for data generation in FedGC framework, we consider the following criteria: generation efficiency, data diversity, and data fidelity. Following the criteria, we explore three crucial aspects, including budget allocation, prompt design, and generation guidance, and propose three representative solutions as candidates for each aspect. Without loss of generality, we use the text-guided latent diffusion model (Rombach et al., 2022) to generate images based on prompts for image task, and ChatGPT OpenAI (2023) to generate texts based on prompts for text task.

**Budget allocation for efficiency.** Though, (1) the process of data generation is just one-shot and (2) FedGC does not compromise on the two first-order concerns in FL: communication cost and privacy (Kairouz et al., 2021), it still costs some computation budget in exchange for algorithm utility (Zhang et al., 2022). Thus, it is essential to design efficient strategies to allocate the generation budget (i.e., the total number of generative samples, denoted as $M$) to each client and label.

To achieve this, we design three allocation strategies. (1) The equal allocation strategy allocates the budget equally to each client and each category, which is the simplest and most general allocation strategy. That is, each client can generate $\frac{M}{KC}$ data samples for each category. (2) Inverse allocation strategy allocates the budget inversely to each client according its number of data samples. Specifically, each client $k$ can generate $\frac{M \cdot (N_{max} - N_k)}{C \cdot \sum_i (N_{max} - N_i)}$ samples for each category, where $N_{max}$ denotes the maximum number in $\{N_i\}_i$. (3) Water-filling-based: each client can generate $\frac{M}{K}$ samples in total, and apply water filling algorithm to allocate samples to each category (Proakis, 2008).

**Prompt design for diversity.** Data diversity plays a key role in learning a generalized model in many domains such as image (Chen et al., 2020) and text (Radford et al., 2019). To increase the diversity, it is essential to design appropriate prompts since they directly guide the process of generation.

For image task, we consider three diversity levels. (1) Single prompt, where we use "a photo of {class}" (Radford et al., 2021). (2) Multiple prompts, where we consider diverse formats such as "{class}". (3) LLM-based diversified prompts, where we instruct an LLM such as ChatGPT to diversify the prompts. While for text generation, we only design one prompt since the ChatGPT (OpenAI, 2023) is sufficient to generate diverse content if we instruct it to be diverse; see Table 8.

**Generation guidance for diversity and fidelity.** Finally, we feed the prompts to the generative models for generation. Besides designing prompts, we randomly set the guidance scale for diffusion models (Rombach et al., 2022) (or non-zero temperature for LLMs) to enhance the data diversity.

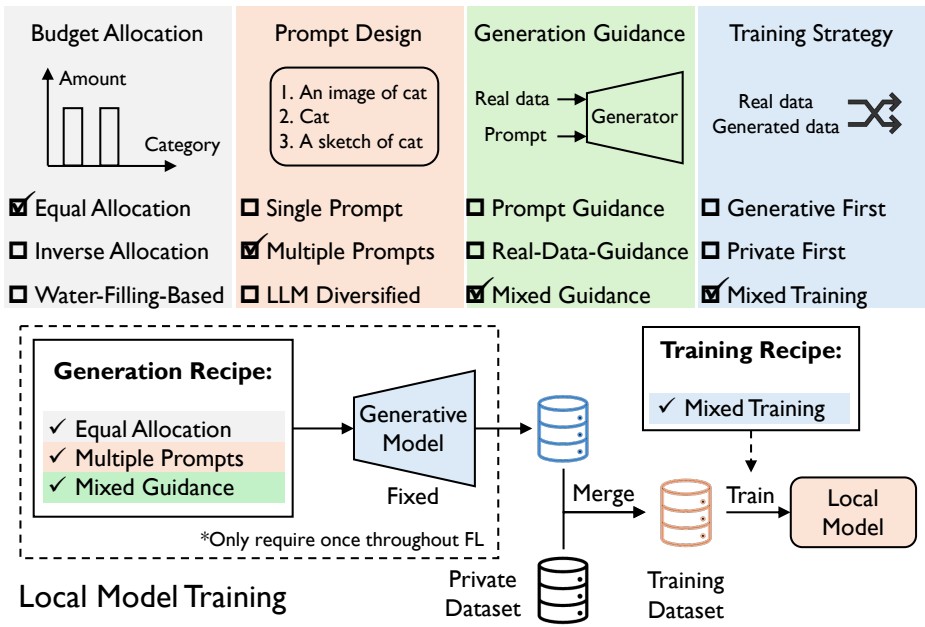

Figure 1: Overview of the designs of FedGC on client side. Above, we summarize four crucial aspects that are worth exploring and propose three solutions for each aspect. Below is the pipeline of local training, where each client first generates data based on the generation recipe, then merges the generative and private dataset, and finally trains the local model based on the training recipe.

(Prompt-Only Guidance) However, data diversity may not be sufficient to ensure improving model training, while data fidelity is also a critical factor. For cases where the domain gap between the generative and real data is too large, the benefits of increasing diversity may be outweighed by the negative effects of the domain gap, leading to degraded performance (He et al., 2022).

(Real-Data Guidance) To alleviate this issue, we propose a new real-data-guided generation approach, which conditions data generation on both real data and prompts. For image task, unlike the original text-guided generation that starts from a random Gaussian noise at latent space $z_T^1$ (Rombach et al., 2022), we propose to inject information of real data into the starting noise. Specifically, we first use the auto-encoder to encode the real image $x$ to latent representation $z$, then add some Gaussian variation to obtain a new $z_T^2$, which substitutes $z_T^1$ as the starting point; see illustration in 7. This enriched latent representation, infused with real data insights, enables the generative model to produce outputs closely resembling real data, optimizing the trade-off between diversity and fidelity. For text task, see illustration in Table 8 using ChatGPT.

(Mixed Guidance) Furthermore, given that certain clients may lack data samples from specific categories, we propose a mixed guidance strategy. Specifically, for a given budget $N_{k,c}$ for client $k$ in category $c$, (1) if client $k$ possesses samples from category $c$, it generates $N_{k,c}/2$ samples using text-only guidance and $N_{k,c}/2$ samples with real-data guidance; (2) in the absence of samples for client $k$ from category $c$, it generates all the $N_{k,c}$ samples using text-only guidance. This approach effectively addresses category omissions and refines the trade-off between diversity and fidelity.

### 3.3 LOCAL MODEL TRAINING IN FEDGC

By choosing generation recipe from the three aspects above, we can generate data using the generative model to assist local model training. Given the generative dataset $\mathcal{D}_g$ and the private dataset $\mathcal{D}_p$, there could be diverse training strategies such as sequential training (optimizing on the two datasets sequentially) and mixed training (optimizing on the mixed dataset).

We find that the mixed training strategy is the most effective despite its simplicity. Thus, we directly merge the two datasets as the final new training dataset $\mathcal{D}_m$, based on which we train the local model

Table 1: Experiments on two heterogeneity types, four datasets, two heterogeneity levels, and six baselines. Test accuracy (%) averaged over three trials is reported. FedGC consistently and significantly brings performance gain over baselines across diverse settings.

| Baseline | H-Type Dataset H-Level | Label Level | | | | Feature Level | | | | Avg. Acc. Δ |
|---|---|---|---|---|---|---|---|---|---|---|
| | | CIFAR-10 | | EuroSAT | | PACS | | VLCS | | |
| | | High | Low | High | Low | High | Low | High | Low | |
| FedAvg | Vanilla | 61.25 | 75.88 | 53.82 | 75.59 | 27.16 | 36.47 | 43.69 | 47.95 | **+12.26** |
| | + FedGC | 74.50 | 79.73 | 74.83 | 84.46 | 54.43 | 53.93 | 46.49 | 50.50 | |
| FedAvgM | Vanilla | 60.83 | 74.40 | 50.91 | 72.80 | 28.96 | 34.52 | 46.64 | 45.74 | **+13.04** |
| | + FedGC | 73.84 | 78.90 | 73.48 | 84.87 | 53.23 | 55.73 | 48.45 | 50.65 | |
| FedProx | Vanilla | 64.02 | 75.62 | 59.61 | 73.20 | 27.71 | 39.52 | 38.83 | 48.50 | **+11.49** |
| | + FedGC | 74.36 | 79.25 | 73.04 | 84.76 | 54.28 | 55.83 | 45.69 | 51.70 | |
| SCAFFOLD | Vanilla | 63.98 | 78.79 | 52.72 | 76.80 | 29.72 | 37.52 | 43.64 | 40.83 | **+12.57** |
| | + FedGC | 73.96 | 80.29 | 69.48 | 81.04 | 59.73 | 60.63 | 47.65 | 51.75 | |
| MOON | Vanilla | 63.40 | 75.43 | 52.67 | 70.02 | 27.91 | 36.52 | 45.89 | 48.30 | **+12.47** |
| | + FedGC | 74.02 | 79.82 | 73.69 | 86.06 | 53.81 | 55.08 | 48.05 | 49.35 | |
| FedDecorr | Vanilla | 64.14 | 76.19 | 63.74 | 69.57 | 27.51 | 29.07 | 37.02 | 47.70 | **+9.62** |
| | + FedGC | 73.94 | 78.16 | 69.93 | 81.30 | 48.77 | 47.42 | 43.39 | 49.00 | |

at the same training manner protocol as other FL methods. Specifically, at the $t$-th FL communication round, each client $k$ first receives the global model $\boldsymbol{\theta}^t$ and re-initializes its local model with $\boldsymbol{\theta}^t$. Then, each client conducts model training based on the merged dataset $\mathcal{D}_m$ for several optimization steps. Finally, each client $k$ obtains its local model $\boldsymbol{\theta}_k^t$, which is subsequently sent to the server for model aggregation ($\boldsymbol{\theta}^{t+1} := \sum_k p_k \boldsymbol{\theta}_k^t$, where $p_k = N_k / \sum_i N_i$ is the relative dataset size).

Note that this process is orthogonal to local training algorithm, which can be SGD-based training (McMahan et al., 2017), proximity-based training (Li et al., 2020b), control-variate-based training (Karimireddy et al., 2020), or feature-alignment-based training (Li et al., 2021).

## 4 EXPERIMENTS

### 4.1 IMPLEMENTATION DETAILS

We set the number of communication rounds as 100. Table 9 lists client number for each dataset.

**Data Heterogeneity and Datasets.** We consider two types of data heterogeneity for image tasks. For label heterogeneity, we consider CIFAR-10 (Krizhevsky et al., 2009) and EuroSAT (Helber et al., 2019), where we allocate the original training dataset to clients based on the frequently used strategy in FL: Dirichlet distribution (Wang et al., 2020a). $\beta$ controls the level of heterogeneity, where we denote 0.05 as high and 0.1 as low. For feature heterogeneity, we consider PACS (Zhou et al., 2020) and VLCS (Fang et al., 2013), where we allocate training dataset of each domain to several clients according to Dirichlet distribution. This captures both the properties of feature- and label-level heterogeneity. For text datasets, we consider Sentiment140 from LEAF benchmark (Caldas et al., 2018) (naturally allocated) and Yahoo! Answers (Zhang et al., 2015) (split by Dirichlet distribution).

**Training Details.** The number of iterations for local model training is 200 and uses SGD as the optimizer with a batch size of 64. The learning rate is set to 0.01 (Li et al., 2021; Ye et al., 2023). We use ResNet-20 (He et al., 2016) for image task and LSTM for text task (Caldas et al., 2018).

### 4.2 EXPERIMENTAL RESULTS

**FedGC significantly improves the FL performance under data heterogeneity.** In Table 1, we show experimental results on two heterogeneity types (label-level and feature-level heterogeneity), two datasets for each type (CIFAR-10, EuroSAT, PACS, and VLCS), and two heterogeneity levels for each dataset. From the table, we see that (1) incorporating baseline in our FedGC framework can consistently and significantly improve the performance of baseline across diverse settings. (2)

FedGC is extremely helpful when the heterogeneity level is relatively high, convincingly supporting our idea of introducing generative data to mitigate the effects of data heterogeneity. Specifically, based on FedAvg, FedGC brings 21.01 absolute accuracy improvement under a high heterogeneity level on EuroSAT and 12.26 absolute accuracy improvement on average.

**FedGC is compatible with existing FL methods.** From Table 1, we see that FedGC consistently and significantly brings performance gain across 6 different baselines, including FedAvg, FedAvgM, FedProx, SCAFFOLD, MOON, and FedDecorr. For example, FedGC averagely brings 12.68 absolute accuracy improvement to SCAFFOLD (Karimireddy et al., 2020). This demonstrates the compatibility and universality of our proposed FedGC framework.

**FedGC achieves better performance and privacy preservation at the same time.** In Figure 2, we show the performance and privacy preservation trade-off comparisons before and after using FedGC. To measure privacy preservation, we use a simple membership inference attack method based on loss evaluation (Yu et al., 2021; Sablayrolles et al., 2019) to evaluate attack accuracy, see details in Section A.3. Lower attack accuracy indicates better privacy preservation. From the figure, we have an interesting finding that our

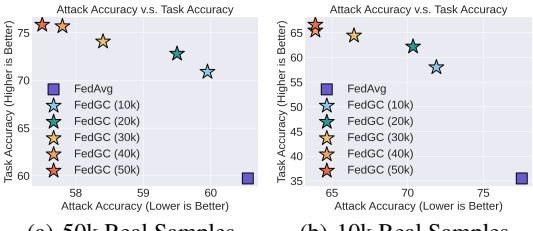

(a) 50k Real Samples    (b) 10k Real Samples

Figure 2: FedGC achieves both better task accuracy and privacy preservation (lower attack accuracy).

FedGC framework can not only improve the performance under data heterogeneity, but also enhance the privacy preservation. We also show in Table 10 that FedGC achieves significantly lower attack accuracy when similar task accuracy is achieved. This is surprising yet reasonable since FedGC requires the model to learn from both the private data and the diverse generative data, meaning that the generative data can dilute the concentration of real, sensitive data.

This explanation can be further verified since (1) as the number of generated samples increases, FedGC achieves lower attack accuracy (better privacy preservation). (2) When the number of real training samples is smaller (from 50k to 10k), we see a much larger reduction in attack accuracy and improvement in task accuracy, since the ratio of private data samples in the whole dataset is lowered.

**FedGC is a general across modalities.** In Figure 3, we report the performance of FedGC in text modality. We consider two datasets, Sentiment140 and Yahoo! Answers, consisting of 1000 and 100 clients, respectively. Here, we use ChatGPT (Ouyang et al., 2022) as the generative model. We apply equal budget allocation and single prompt (where we increase the diversity by directly instructing ChatGPT to "be diverse"). For real-data-guidance, we take advantage of LLM's few-shot learning ability by giving several real examples in the context (Brown et al., 2020). From the figure, we see that FedGC still consistently and significantly

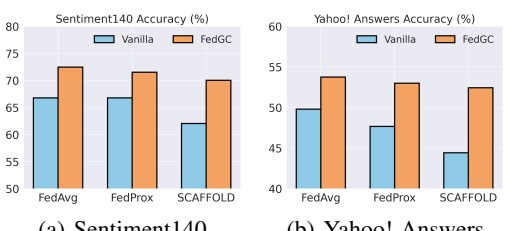

(a) Sentiment140    (b) Yahoo! Answers

Figure 3: Performance comparisons on two text datasets. Our proposed FedGC consistently and significantly brings performance gain.

brings performance gain to all three baselines across two datasets. This experiment verifies that our proposed FedGC framework has the potential to generalize well to diverse modalities.

**Applicability to diverse scenarios.** We also (1) consider scenarios where only a partial of the clients are capable of generating data in Section A.6; (2) experiment on partial client participation scenarios in Section A.8; (3) experiment under different heterogeneity levels in Section A.7.

### 4.3 TOWARDS DIFFERENT DESIGNS OF FEDGC

**Generating more data could make FedAvg prevail.** In Table 2, we explore the effects of number of generated samples on FL's performance. 0 denotes vanilla FL baseline. Experiments are conducted on CIFAR-10 ($\beta = 0.05$). From the table, we have an interesting finding: (1) when the number of generated samples is relatively small (0∼2000), FedGC can enlarge the gap between standard FedAvg and the method (SCAFFOLD) that is specifically designed for addressing data heterogeneity;

Table 2: Increasing number of generated samples makes FedAvg (McMahan et al., 2017) prevail.

| No. Gen. | 0 | 100 | 200 | 500 | 1000 | 2000 | 5000 | 10000 | 20000 | 50000 |
|---|---|---|---|---|---|---|---|---|---|---|
| FedAvg | 61.25 | 63.67 | 66.21 | 67.13 | 66.98 | 66.28 | 71.65 | **74.50** | **76.93** | **76.39** |
| FedProx | **64.02** | 66.47 | 67.40 | 67.05 | 68.55 | 69.19 | **72.10** | **74.36** | **76.81** | **76.73** |
| SCAFFOLD | **63.98** | **69.05** | **71.33** | **71.55** | **71.33** | **70.04** | 70.34 | 73.96 | 74.88 | 73.98 |

Table 3: Different prompt designs of FedGC applied on baselines. The design of multiple prompt formats is preferred for its effectiveness, diversity, and simplicity.

| Baseline | No-GC | Single | **Multiple** | LLM |
|---|---|---|---|---|
| FedAvg | 27.06 | 50.53 | **54.08** | 41.32 |
| FedProx | 29.12 | 50.48 | **53.03** | 40.82 |
| SCAFFOLD | 28.56 | 54.13 | **58.53** | 45.87 |

Table 4: Different generation guidance designs of FedGC applied on baselines. The mixed guidance that combines text2img and img&text2img is the most effective strategy.

| Baseline | Pri. | T2I | IT2I | **Mixed** |
|---|---|---|---|---|
| FedAvg | 48.57 | 51.91 | 42.38 | **56.67** |
| FedProx | 49.52 | 51.43 | 44.76 | **56.19** |
| SCAFFOLD | 54.76 | 56.67 | 49.52 | **58.57** |

Table 5: Different budget allocation strategies of FedGC applied on baselines. Equal allocation is preferred for its effectiveness and simplicity.

| Baseline | **Equal** | Inverse | Water |
|---|---|---|---|
| FedAvg | **74.50** | 68.10 | 71.26 |
| FedProx | **74.36** | 68.51 | 72.23 |
| SCAFFOLD | 73.96 | 73.94 | **74.43** |

Table 6: Different training strategies of FedGC applied on baselines. Generated data can only exhibit its efficacy when used in conjunction with real data. **Mixed** training is the most effective.

| Baseline | Pri. | Gen. | P2G | G2P | **Mixed** |
|---|---|---|---|---|---|
| FedAvg | 60.77 | 41.85 | 67.06 | 67.11 | **73.99** |
| FedProx | 63.62 | 40.93 | 67.23 | 69.04 | **73.69** |
| SCAFFOLD | 65.00 | 43.45 | 66.73 | 69.50 | **75.79** |

(2) however, as the number continues to grow, the situation is reversed that the basic FL method FedAvg prevails. This finding suggests that apart from carefully designing FL algorithm, it is also a promising direction to explore the greater potential from the perspective of generative data.

**Equal allocation is a preferred allocation strategy for its effectiveness and simplicity.** In Table 5, we compare different budget allocation strategies on CIFAR-10, including equal allocation, inverse allocation, and water-filling-based allocation. Experiments show that equal allocation contributes to better performance for both FedAvg and FedProx, and comparable performance compared with water-filling-based allocation for SCAFFOLD. Considering effectiveness and simplicity, we conclude that equal allocation is a preferred allocation strategy.

**Multiple prompts lead to better performance, while LLM-based diversification might be unnecessary.** In Table 3, we explore different prompt designs on PACS dataset. PACS contains significant label-level and feature-level variations, making it an apt choice for this exploration. We compare baseline without FedGC, FedGC with single, multiple, and LLM-based prompts (see prompt generation in Table 7). From the table, (1) we see that FedGC incorporated with all the prompt designs improves the performance of baselines (see improvement over the No-GC column). (2) We see that multiple prompts consistently and significantly perform better, while LLM-based prompts perform ordinarily. This may result from the fact that the scene descriptions from the LLM are usually complicated, causing multifaceted patterns in one sample, thereby complicating model training. Overall, we prefer the design of multiple prompts for its effectiveness, diversity, and simplicity.

**Mixed guidance contributes to higher performance for rare tasks.** In Table 4, we compare different generation guidance designs on a medical dataset HAM10000 (Tschandl et al., 2018). The reason for choosing this dataset is that the diffusion model (Rombach et al., 2022) fails to correctly understand medical prompts (Kazerouni et al., 2022), which helps support our claim more convincingly. We consider three designs, including text-guided generation (T2I), our proposed data generation with guidance of text and real data (IT2I), and the mixed usage of T2I and IT2I. These

experiments convey three interesting findings: (1) even though the diffusion model fails to generate data that visually agrees with real data, the generated data still contributes to enhancing the performance of FL (see improvement from Pri. to T2I). (2) IT2I itself fails to bring performance gain, which may result from the limited diversity and incapability to generate for missing classes. (3) Mixing these two strategies contributes to consistently and significantly better performance.

**Mixed training is the most effective training strategy.** In Table 6, we compare different training strategies on CIFAR-10, including training only on the private dataset (Pri.), training only on the generative dataset (Gen.), sequential training with private dataset first (P2G), sequential training with generative dataset first (G2P), and mixed training. Experiments show that 1) generative data itself fails to ensure training, indicating that there is a gap between generative data and real private data. 2) However, when using generative data together with real private data, we see consistent performance gain compared to training on private data. This indicates that despite the incapability of fully representing real data, the generative data still contributes to improving training by increasing diversity. 3) Mixed training consistently and significantly achieves better performance.

## 4.4 Towards Deeper Understanding of FedGC

**Generated data is diverse, but may not be similar to real data.** In Figure 8, we visualize the real data and generated data on EuroSAT. We notice that the generated data samples do not always closely resemble real images, indicating the gap between generative data and real private data (at least visually). Yet, their inclusion still improves the FL's performance under data heterogeneity, which may result from two perspectives. (1) The generative data might act as a form of data augmentation, which potentially introduces variations that are not covered by the original dataset. (2) The generative data diversify the dataset, which serves as a form of implicit regularization, preventing the model from over-fitting to the potentially biased private local data.

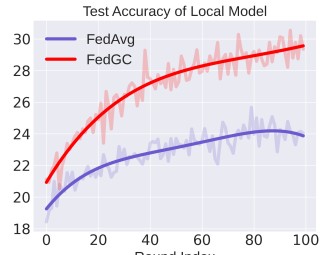

Figure 4: Local models in FedGC better preserve capability in general tasks.

**FedGC alleviates over-fitting local data distribution.** In Figure 4, we compare the averaged test accuracy of local models on the global test dataset. From the figure, we can see a clear accuracy gap between our FedGC and the baseline FedAvg. (1) This indicates that our proposed FedGC can encourage each client to preserve the capability on the global general task, rather than overly fit the local specific task (local data distribution). (2) This also helps explain why the generative data can bring performance gain even though they may fail to resemble real data.

**FedGC reduces data heterogeneity.** In Figure 5, we explore the effects of FedGC on data heterogeneity from the perspective of data. To measure the data heterogeneity, we first extract the features of data for each client using a pre-trained ResNet-18 (He et al., 2016), average the features, and compute the pair-wise cosine similarity among the averaged features of all clients. Figure 5 shows the pair-wise similarity using Client 9 as the reference. From the figure, we consistently see that FedGC can significantly increase the similarity between datasets of two clients, verifying that FedGC can contribute to mitigating data heterogeneity. We also report $\ell_2$ distance as metric and results on PACS in Figure 9 and 10.

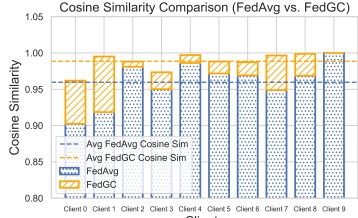

Figure 5: FedGC increases cosine similarity between local datasets (i.e., reduces data heterogeneity).

**FedGC implicitly reduces model divergence.** In Figure 6, we visualize the local model divergence along with the round increases. Specifically, at each round, we compute the $\ell_2$ difference between each local model and the aggregated global model (Li et al., 2020b) and report the averaged difference. (1) From Figure 6(a), we see that FedGC consistently and significantly reduces the model divergence of local models under severe heterogeneity level ($\beta = 0.05$). This result well supports the claim that FedGC is a pleasant FL framework for tackling the issue of data heterogeneity since it has been shown that data heterogeneity leads to larger model divergence and thus mediocre performance empirically (Li et al., 2020b) and theoretically (Wang et al., 2021; Li et al., 2020a).

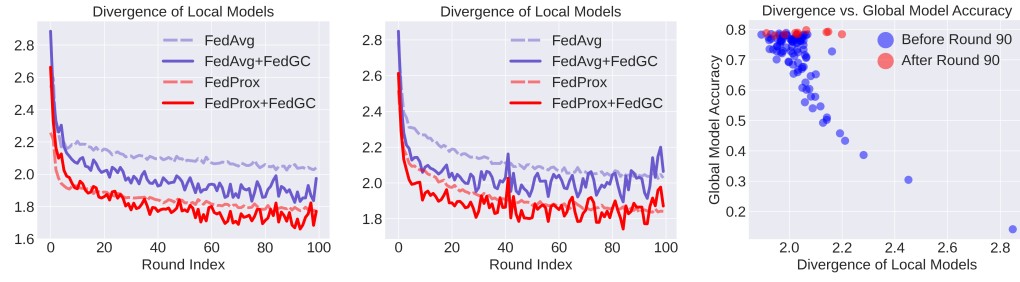

(a) Model Divergence at $\beta = 0.05$   (b) Model Divergence at $\beta = 0.1$   (c) Divergence v.s. Accuracy

Figure 6: FedGC implicitly reduces model divergence. Notably, even with higher divergence in final rounds, we still observe higher accuracy, reflecting the potential dual nature of data diversity.

(2) From Figure 6(b), though the divergence of FedGC increases at the final rounds, we still observe improved accuracy during these rounds. This observation is interesting as it seems to contradict the current viewpoint. Based on this, we hypothesize that data diversity has two sides: (1) it can reduce data heterogeneity, thus reducing model divergence; (2) but it can also increase model divergence as the data is more diverse. Nevertheless, data diversity still contributes to enhanced model performance despite the increased divergence as suggested in Figure 6(c) (see the red scatters). This interesting finding calls for more theoretical future works to model data diversity rather than only model divergence in the FL theory (Karimireddy et al., 2020; Wang et al., 2020b).

## 5   DISCUSSIONS AND FUTURE DIRECTIONS

As an initial exploration of tackling data heterogeneity in FL using generative content, we mainly focus on designing under two standards: simplicity and effectiveness. However, there are also diverse interesting future directions that are worth exploring.

**Filtering.** Previously, we find that the positive effects of data diversity on data heterogeneity outweigh the negative effects of some low-quality generated samples and the gap between generated and real data. Still, it could benefit if we can propose to appropriately filter out low-quality samples, such as applying the KNN method (Fix & Hodges, 1989) using real data as reference. We propose an initial attempt using the global model as a discriminator to filter data in Section A.9.

**Personalized generation.** It would also be advantageous to narrow the gap between generated and real data. Beyond our proposed real-data-guided generation, delving into personalized data generation to more closely resemble real data is worth investigating. For example, we can fine-tune generative models on real data using parameter-efficient fine-tuning techniques (Hu et al., 2021).

**Theory.** As shown in Figure 6(b) and 6(c), model divergence (Li et al., 2019; 2020b; Wang et al., 2021) may not fully represent the property of distributed data, calling for more future theory works.

## 6   LIMITATIONS AND CONCLUSION

*Limitations and Future Works.* Despite putting much effort into diversifying the experimental settings, there are still cases not covered. For example, we only explore one diffusion model and LLM respectively. There could be future works to explore the effects of different generative models.

*Conclusion.* This paper focuses on the notorious issue of data heterogeneity in FL. We propose a new, simple yet effective FL framework termed FedGC, which leverages generative data to promote FL under heterogeneous private data. We summarize four crucial aspects that are worth exploring and propose three solutions for each aspect. Extensive experiments show that our FedGC framework can consistently and significantly improve the performance of diverse FL baselines under data heterogeneity. Moreover, we provide a systematic empirical analysis based on FedGC and provide new insights throughout the experimental section. Our research serves as an initial exploration of boosting federated learning on private data in the era of generative content.

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

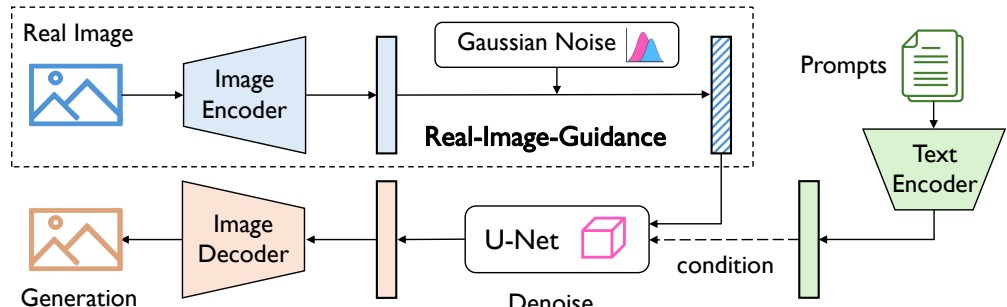

Figure 7: Real-data-guidance for image generation based on diffusion model. The real-data-guidance method involves 4 steps: (1) initializing latent features with real-image data, (2) adding controlled noise, (3) denoising with text features, and (4) generating new images using the decoder.

## A   APPENDIX

### A.1   MORE ILLUSTRATION OF FEDGC

For the prompts conditioned on the latent diffusion model, we show the LLM-based prompts for generating images in Table 7. In detail, we instruct ChatGPT through System Prompt and User Prompt, to help us create text samples containing the corresponding class name for image generation. Utilizing ChatGPT's rich imagination of scenarios and the diversity of text styles, we can achieve a diversity of prompts. Therefore, it helps Stable-diffusion to generate diverse and more realistic pictures.

For generation guidance beyond prompts, we show the real-data guidance for image generation using diffusion models in Figure 7. First of all, the latent features are meticulously initialized using actual real-image data. Subsequently, controlled noise is introduced into the latent representations, which serves to perturb and diversify the features while maintaining the underlying structure. Following this, with conditioned prompts, we denoise this combined feature using U-Net (Ronneberger et al., 2015). Finally, passing through the image decoder, we obtain generated images.

We show the real-data-guidance for text generation using ChatGPT in Table 8. Compared to prompts containing class num, here we instruct ChatGPT to imitate the theme and content of the corresponding text and directly expand the amount of text data. In our illustrative examples shown in Table 8, we simulate real-world data scenarios by incorporating four actual instances and generating an additional set of four synthetic instances. In this experimental setup, we task ChatGPT with the generation of data that exhibits diverse patterns akin to those found in authentic real data. Furthermore, we guide ChatGPT to produce two distinct samples for each distinct label category, fostering a balanced and representative dataset.

### A.2   IMPLEMENTATION DETAILS

We list the number of clients for each dataset in Table 9.

### A.3   MEMBERSHIP INFERENCE ATTACK

To measure the privacy preservation of FedAvg and FedGC, we carry out a simple membership inference attack based on loss evaluation, as (Sablayrolles et al., 2019) has shown that it is reasonable to use the loss of the model to infer membership. We consider a scene where an attacker who has a tiny amount of training data can get the global model and wants to figure out whether a similar datum (i.e. also a photo of an airplane) has been used to train the model or not. During the attack, the attacker feeds its few data to the global model and trains a binary classifier based on the loss of each training-used and not-training-used datum.

We conduct our experiment on CIFAR-10 dataset. In the training process, we set the client number to 10 and the Dirichlet distribution parameter to $\beta = 0.1$. We also discard data augmentations

Table 7: Obtaining LLM-based prompts for generating images using diffusion models. Instructions for generating scene descriptions (i.e., prompts for diffusion models) given a class name using ChatGPT. Here, we provide an example on the dog category of PACS dataset.

---

**System Prompt:**
You are an AI assistant that helps people find information.

**User Prompt:**
Please help me come up with scene descriptions that contain a dog while not containing an elephant, giraffe, guitar, horse, house, person.

For example:

["A dog is running on the grass", "A dog is sleeping on the floor"]

Please generate 10 samples in the format of a list.
Remember: each description should be within 10 words.

---

Table 8: Real-data-guidance for text generation using ChatGPT. Real data is modeled in the examples, where we provide four real examples and generate four new examples. We instruct the ChatGPT to generate diverse data that has a similar pattern to real data. We also instruct the ChatGPT to generate two samples for each label.

---

**System Prompt:**
Assistant is an intelligent chatbot designed to help users generate similar data. Users will provide a few real samples and the Assistant will generate data that follows the pattern of real samples. This is a binary dataset on sentiment analysis, where 0 denotes negative and 1 denotes positive.

Instructions:
1. Generate two samples with label 0 and two samples with label 1, try to make the content diverse
2. Should have a similar pattern of users' data.

**User Prompt:**
**Data: {example_input_1}, Label: {example_label_1}**
**Data: {example_input_2}, Label: {example_label_2}**
**Data: {example_input_3}, Label: {example_label_3}**
**Data: {example_input_4}, Label: {example_label_4}**

Generate two samples with label 0 and two samples with label 1.
In the format of Data: {}, Label: {}. Each sample should start with ** and end with **.

---

(i.e. flipping and cropping) for more clear comparisons. In the main body, we compare both task accuracy and attack accuracy, as shown in Figure 2.

We also compare the attack accuracy at the point when FedAvg and FedGC achieve similar task accuracy in Table 10. From the table, we see a much more significant reduction in privacy leakage (i.e., much lower attack accuracy). This is reasonable as FedGC can accelerate the convergence speed, which means FedGC requires fewer steps of optimization on the sensitive private data to achieve the same.

Table 9: Number of clients for each dataset.

| Dataset | CIFAR-10 | EuroSAT | PACS | VLCS | HAM10000 | Sentiment | Yahoo! |
|---|---|---|---|---|---|---|---|
| Client Number | 10 | 10 | 20 | 20 | 10 | 1000 | 100 |

Table 10: Membership inference attack accuracy comparisons when FedAvg and FedGC achieve similar task accuracy. We consider two scenarios where the total number of clients' real samples is 50k and 10k, respectively. We also explore the effects of using different number of generated samples. FedGC can reduce privacy leakage to a very low level (since random guess is 50%) while maintaining task accuracy at the same time.

| Number of Real Samples Accuracy | | 50k | | 10k | |
|---|---|---|---|---|---|
| | | Task | Attack | Task | Attack |
| | 0 | 59.71 | 60.55 | 35.48 | 77.55 |
| | 10k | 61.65 | 52.05 | 35.97 | 52.80 |
| No. of Generated Samples | 20k | 62.49 | 51.20 | 39.18 | 52.85 |
| | 30k | 61.82 | 51.95 | 39.40 | 52.50 |
| | 40k | 60.38 | 51.20 | 37.17 | 52.75 |
| | 50k | 62.49 | 51.60 | 38.68 | 52.35 |

## A.4 VISUALIZATION OF REAL AND GENERATED DATA

We visualize the real data and generated data on EuroSAT (Helber et al., 2019) in Figure 8. For the uncommon and detailed satellite images in EuroSAT (Helber et al., 2019), the quality of the data generated by the diffusion models varies. From the naked eye, the data generated by some diffusion can capture the semantic information brought by the label very well. For example, the generated images with the label "River" as guidance do contain rivers, but hard to achieve a similar satellite style to actual images. Although the gap between generated and actual data definitely exists, generated data obviously improves specific task performance, which is demonstrated by our extensive experiments.

## A.5 FEDGC MITIGATES DATA HETEROGENEITY

We visualize the cosine similarity and $\ell_2$ distance of features on EuroSAT and PACS in Figure 9 and Figure 10 respectively. We measure the discrepancy among local data in clients on the feature level, using 2 metrics: cosine similarity and $\ell_2$ distance. To be specific, we calculate the average features with pre-trained ResNet-18 (He et al., 2016) on each client in turn, and then measure the indicators between all pairs of clients.

Results in the figures manifest that after applying FedGC, the cosine similarity and $\ell_2$ distance among client pairs separately increase and decrease. In other words, local data possessed by clients are more homogeneous than before. FedGC efficiently mitigates data heterogeneity by generating corresponding data on the client side. From the feature respective, we show the latent reason for significant performance improvement brought by FedGC.

## A.6 FEDGC WITH PARTIAL CLIENTS CAPABLE OF GENERATION

Our proposed FedGC framework is also applicable in cases where not every client has the capability to generate data. Here, we experiment on CIFAR-10 under two different heterogeneity levels. In Table 11, we compare vanilla baseline with no generative data, FedGC where all clients can generate data, and FedGC where only half of the clients can generate data.

From the table, we see that (1) our proposed FedGC can consistently and significantly achieve the best performance despite the amount of generation-capable clients. (2) Surprisingly, we find that under low heterogeneity level, when applied to SCAFFOLD (Karimireddy et al., 2020), FedGC

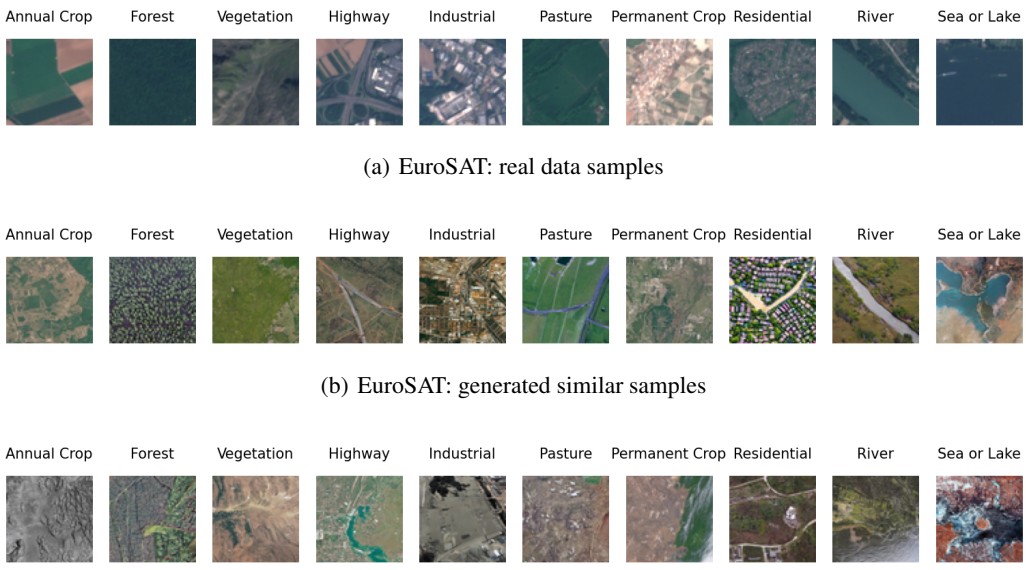

Figure 8: Visualization of real and generated data. (a) Visualization of real data samples from the EuroSAT dataset. (b) Visualization of generated data samples that are more aligned with the corresponding semantic or real data. (c) Visualization of generated data samples that are not aligned with the corresponding semantic or real data.

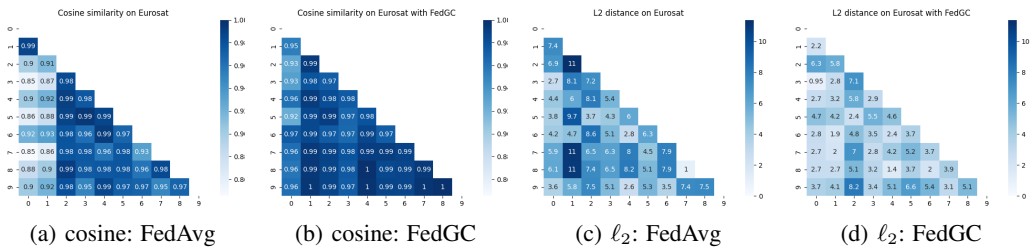

Figure 9: Feature cosine similarity and $\ell_2$ distance heatmap among 10 clients on EuroSAT. We calculate the two metrics on average data features among clients using the pre-trained ResNet-18 (He et al., 2016). FedGC enhances the feature similarity and closes their distance, which effectively mitigates the feature-level heterogeneity on EuroSAT.

with few generation-capable clients even performs better. This interesting finding demonstrates that our framework may be further improved by more fine-grained designs regarding who is responsible for data generation and the volume of data to be generated.

## A.7 FEDGC UNDER DIFFERENT HETEROGENEITY LEVELS

Here, we conduct experiments of three baselines including FedAvg, FedProx, and SCAFFOLD, with different heterogeneity levels on CIFAR-10. The Beta $\beta$ stands for the hyper-parameter in the Dirichlet distribution. As $\beta$ increases in [0.05, 0.07, 0.1, 0.3, 0.5, 1.0, 5.0], the data heterogeneity level reduces. Illustrated in Figure 11, we can observe that (1) FedGC consistently outperforms these three algorithms in all different data heterogeneity levels. (2) As the heterogeneity level increases, the accuracy improvement brought by FedGC significantly elevates, which showcases the reliability of FedGC to mitigate heterogeneity, one of the intricate issues in FL.

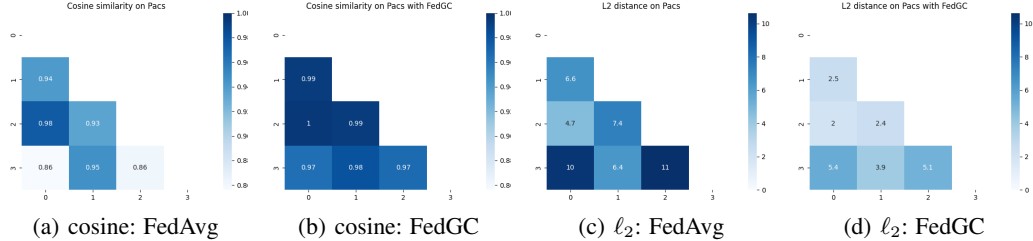

|            | (a) cosine: FedAvg | (b) cosine: FedGC | (c) $\ell_2$: FedAvg | (d) $\ell_2$: FedGC |

Figure 10: Feature cosine similarity and $\ell_2$ distance heatmap among 4 clients on PACS. We calculate the two metrics on average data features among clients using the pre-trained ResNet-18. FedGC enhances the feature similarity and closes their distance, which effectively mitigates the feature-level heterogeneity on PACS.

Table 11: Experiments of a scene in which partial clients are capable of generation. 1k/50% indicates only half of the clients are capable of generation. However, FedGC still significantly outperforms the baseline with no generative data.

| H-Level Generation | High | | | Low | | |
|---|---|---|---|---|---|---|
| | No | 1k/100% | 1k/50% | No | 1k/100% | 1k/50% |
| FedAvg | 60.77 | 73.99 | 71.53 | 71.57 | 79.73 | 77.45 |
| FedProx | 63.62 | 73.69 | 72.65 | 75.76 | 79.25 | 79.23 |
| SCAFFOLD | 65.00 | 75.75 | 73.28 | 78.74 | 80.29 | 81.27 |

## A.8 FEDGC FOR PARTIAL CLIENT PARTICIPATION SCENARIOS

Here, we conduct experiments of three baselines including FedAvg, FedProx, and SCAFFOLD on CIFAR-10 with Dirichlet distribution parameter $\beta = 0.1$. Specifically, we set the communication round to 200, local iteration number to 100, and try different client number and participation rate. As illustrated in Table 12, we can observe that FedGC still significantly outperforms the baseline with no generated data under each circumstance.

## A.9 GLOBAL-MODEL-BASED DATA FILTERING

We propose global-model-based data filtering, where each client conducts data filtering on the client side according to the received global model before local model training. Specifically, to determine which data to filter, a client feeds its generated data to the global model to evaluate the loss value for each data sample. Then, each client selects the top $x\%$ data (we set $x = 90$ here) and mixes the selected generated data with its real data.

Furthermore, since the global model might perform drastically differently on different categories, simply selecting according to the loss of all data samples may result in imbalanced filtering. That is, this could make to global model filter out most of the samples where it performs poorly. Addressing this, we further propose category-wise data filtering based on global model, which filers the same ratio of data for each category.

Here, we perform experiments on EuroSAT dataset with two heterogeneity levels in Table 13. Vanilla denotes FedAvg itself, No F denotes FedGC without filtering, F@50 denotes filtering from round 50, F@50-C denotes category-wise filtering. From the table, we see that (1) under a high heterogeneity level, F@75 contributes to higher performance than No F, even with only 90% of data at final rounds. (2) Category-wise filtering generally performs better than unified filtering, indicating its effectiveness. (3) Nevertheless, such filtering technique can not always ensure performance improvement, calling for more future work. The performance drop could result from reduced number of data samples and ineffective filtering.

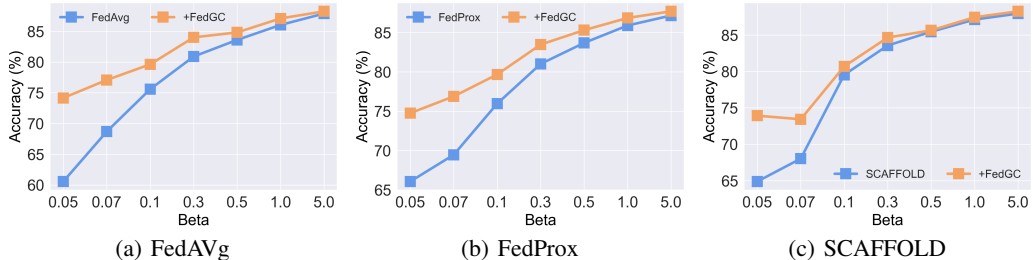

|              | (a) FedAVg | (b) FedProx | (c) SCAFFOLD |
| :---: | :---: | :---: | :---: |

Figure 11: Performance comparisons between vanilla baseline and baseline in FedGC framework under different heterogeneity levels on CIFAR-10. Beta ($\beta$) is the hyper-parameter in Dirichlet distribution. As the heterogeneity level increases (Beta decreases), the improvement brought by FedGC becomes more significant. This indicates that FedGC can effectively alleviate the issue of data heterogeneity.

Table 12: Experiments of a scene in which only partial clients participate in training each round. We conduct experiments on three different total client numbers and several different participation rates. For example, client 200 and participation rate 5% means randomly selecting 10 clients to participate in training each round. In each case, FedGC still significantly outperforms the baseline with no generative data.

| Baseline | Client Participation | 200 | | | 100 | | 50 | |
| :---: | :---: | :---: | :---: | :---: | :---: | :---: | :---: | :---: |
| | | 5% | 10% | 20% | 10% | 20% | 10% | 20% |
| FedAvg | Vanilla | 53.62 | 60.00 | 65.76 | 56.53 | 57.69 | 55.90 | 63.33 |
| | + FedGC | 68.93 | 74.06 | 75.74 | 74.16 | 74.26 | 75.34 | 77.20 |
| FedProx | Vanilla | 53.93 | 59.95 | 64.53 | 56.74 | 59.54 | 56.36 | 65.66 |
| | + FedGC | 70.23 | 73.79 | 75.07 | 74.39 | 74.05 | 75.47 | 77.47 |
| SCAFFOLD | Vanilla | 60.41 | 68.02 | 70.15 | 65.03 | 68.12 | 65.73 | 72.42 |
| | + FedGC | 71.65 | 74.83 | 77.54 | 74.38 | 76.26 | 72.74 | 77.56 |

Table 13: Experiments of global-model-based data filtering. We conduct our initial attempt on EuroSAT dataset with two heterogeneity types. F@50 means start filtering after 50 communication rounds and C means filtering by each class.

| Heterogeneity Level | Vanilla | No F | F@50 | F@75 | F@50-C | F@75-C |
| :---: | :---: | :---: | :---: | :---: | :---: | :---: |
| High | 53.82 | 74.83 | 72.96 | 74.93 | 73.50 | 74.20 |
| Low | 75.59 | 84.46 | 83.82 | 83.83 | 84.19 | 83.83 |

Overall, here we just provide an initial attempt to consider the potential of data filtering. We believe more future works could be proposed to better filter the generated data such that we could use the generated data more efficiently.

