# OpenReview forum: "Federated Learning Empowered by Generative Content"
_ICLR.cc/2024/Conference — Submitted to ICLR 2024_

### Official Review · Reviewer_kS7r · 2023-10-23

**Soundness:** 3 good
**Presentation:** 3 good
**Contribution:** 3 good
**Rating:** 8
**Confidence:** 4

**Summary:**

This paper proposes an idea of adding generated data to local client datasets in federated learning to improve local model performances. Coined under the name FedGC but the framework is basically the standard federated learning framework plus data generation for local clients. In data generation, the paper focuses on 4 aspects: budget allocation, prompt design, generation guidance an training strategy. For each aspects, the authors propose 3 simple approaches.

To show the superiority of FedGC experimentally, they created a dataset with synthesized heretogeneity by merging from a few existing datasets: CIFAR-10, EuroSAT, PACS, VLCS, Sentiment140 from LEAF benchmark and Yahoo! Answers. They tested with a few federated learning frameworks including FedAvg, FedProx and SCAFFOLD and showed that with data generation, the models performed better.

**Strengths:**

The flow of the paper is clear, straightforward and easy to read. The motivation of the problem is exciting.

**Weaknesses:**

There are a few issues in the paper.

1. Other than considerations regarding generating data locally for each client (i.e. the four aspects at generating the data above), there is no significant theoretical contribution. There is no theorem, no proposal. Not a single equation is found in the paper.

2. The paper solely focuses on data generation for local training. However, there is nothing in the communication among the clients that carries any information about data generation from one client to another, other than the number of samples to be generated for each client. In other words, the use of federated learning and data generation appear to be unrelated. I wonder if the whole work could have been better presented in a non-federated setting.

3. In terms of data generation for each client itself, the proposed approaches for each of the 4 aspects above are simple and straightforward. I think this part of the paper is good from a practical point of view. However, it appear to be dominantly engineering contributions, which does not seem suitable for ICLR, a conference about learning representations.

4. The dataset for experimenting was made up by merging from a few existing known datasets which were not designed for federated learning. Heterogeneity in the dataset was synthesized using Dirichlet distribution. This is probably one major weakness of the paper. It is questionable whether the dataset reflects real world. Results in this dataset are therefore not very convincing. I reckon the authors to use real-world datasets, or if that task is not feasible, at least use the same datasets that other federated learning approaches have used, rather than creating one of your own.

**Questions:**

FedGC seems to be a straightforward idea of using existing generative models to generate data for learning. Doesn't that mean a related generative model has to exist for a given problem? What if no generative model exists for a given problem?

---

> ### Author Response · Authors · 2023-11-21
>
> Thank you for your time and comments.
>
>
> **General Response:** We think that the main reason of the reviewer's rejection is the seemingly simplicity of our method. However, we would like to justify for ourselves.
>
> 1. We effectively mitigate the issue of one of the most critical issuses in FL: data heterogeneity. Our method is simple yet effective as verified by the **consistent and significant performance improvement**, which should not be overlooked.
>
> 2. Simplicity does not equal no novelty. Our proposed FedGC framework for the **first time** training private heterogeneous data mixed with synthetic data generated by the powerful advanced generative models. Our work points out its potential and we believe that it could inspire more explorations in this topic to better address or even fully address the issue of data heterogeneity.
>
> 3. We intentionally to make our framework simple and general. As the first exploration on this topic, rather than designing seemingly complicated algorithm, we decided to implement a general framework to conduct a **systematic empirical study** to provide more fundamental insights. Besides, thanks to this simplicity, our method is naturally compatible with conventional FL communication protocols such as secure aggregation and differential privacy.
>
> ---
>
> **W2:** The paper solely focuses on data generation for local training. However, there is nothing in the communication among the clients that carries any information about data generation from one client to another, other than the number of samples to be generated for each client. In other words, the use of federated learning and data generation appear to be unrelated. I wonder if the whole work could have been better presented in a non-federated setting.
>
> **Response:** Focusing on data heterogeneity, we propose a new method that can effectively alleviate its negative effects. The generative data can significantlly mitigate the level of data heterogeneity and the issue of overfitting, promoting the performance of FL. Thus, it is not 'unrelated'.
>
> Data heterogeneity is a representative and common issue in federated setting while in a non-federated setting there is no defination of data heterogeneity. Thus, the idea of using generative content to address data heterogeneity is quite **unique and suited for FL** but not non-FL setting. From the following table, we can see that local training (without FL) with generative content performs significantly worse.
>
> [**Table R1.** Performance comparison between local training with generative content and our FedGC.]
> | Method | CIFAR-High | CIFAR-Low | EuroSAT-High | EuroSAT-Low |
> |:--------:|:----------:|:---------:|:------------:|:-----------:|
> | Local+GC |   46.89    |   50.47   |    24.87     |    35.48    |
> |  FedGC   | **74.50**  | **79.93** |  **74.83**   |  **84.46**  |
>
>
> To make the design 'more like' FL, there could be many methods following FedGC framework. For example, to mitigate the potential negative effects of dissimilarity between generative and real data, we can use soft-label supervision to train local model on generative data uwhile hard-label supervision for real data, where the soft labels come from the FL global model. We leave such methods to future works.
>
> ---
>
> **W1/W3:** There is no theorem, no proposal. Not a single equation is found in the paper. I think this part of the paper is good from a practical point of view. However, it appear to be dominantly engineering contributions, which does not seem suitable for ICLR, a conference about learning representations.
>
> **Response:**
>
> First, federated learning itself is a practical setting and we propose a practical solution, which naturally fits the overall objective.
>
> Second, this judgment varies from person to person. We kindly request the reviewer to refer to the following ICLR papers, where they might seem to be 'unrelated' and 'engineering' for the reviewer.
>
> [1] Chen, Hong-You, et al. "On the importance and applicability of pre-training for federated learning." The Eleventh International Conference on Learning Representations. 2023.
>
> [2] Nguyen, John, et al. "Where to Begin? On the Impact of Pre-Training and Initialization in Federated Learning." The Eleventh International Conference on Learning Representations. 2023.
>
> [3] Oh, Jaehoon, SangMook Kim, and Se-Young Yun. "FedBABU: Toward Enhanced Representation for Federated Image Classification." International Conference on Learning Representations. 2022.

---

> ### Author Response · Authors · 2023-11-21
>
> **W4:** The dataset for experimenting was made up by merging from a few existing known datasets which were not designed for federated learning?
>
> **Response:**
>
> Please note that most of the used datasets are **quite common in federated learning literature and we follow exactly the same setting. Here are some examples.**
>
> - Sentiment140, which is from FL LEAF benchmark [1] (1052 citations).
> - CIFAR10, which is commonly used in most of FL papers [2,3,4].
> - HAM10000 [5,6,7], PACS[8,9,10].
>
> [1] Caldas, Sebastian, et al. "Leaf: A benchmark for federated settings." arXiv preprint arXiv:1812.01097 (2018).
>
> [2] Chen, Hong-You, et al. "On the importance and applicability of pre-training for federated learning." The Eleventh International Conference on Learning Representations. 2023.
>
> [3] Nguyen, John, et al. "Where to Begin? On the Impact of Pre-Training and Initialization in Federated Learning." The Eleventh International Conference on Learning Representations. 2023.
>
> [4] Oh, Jaehoon, SangMook Kim, and Se-Young Yun. "FedBABU: Toward Enhanced Representation for Federated Image Classification." International Conference on Learning Representations. 2022.
>
> [5] Thapa, Chandra, et al. "Splitfed: When federated learning meets split learning." Proceedings of the AAAI Conference on Artificial Intelligence. Vol. 36. No. 8. 2022.
>
> [6] Yang, Fu-En, Chien-Yi Wang, and Yu-Chiang Frank Wang. "Efficient model personalization in federated learning via client-specific prompt generation." Proceedings of the IEEE/CVF International Conference on Computer Vision. 2023.
>
> [7] Pennisi, Matteo, et al. "Experience Replay as an Effective Strategy for Optimizing Decentralized Federated Learning." Proceedings of the IEEE/CVF International Conference on Computer Vision. 2023.
>
> [8] Zhang, Ruipeng, et al. "Federated domain generalization with generalization adjustment." Proceedings of the IEEE/CVF Conference on Computer Vision and Pattern Recognition. 2023.
>
> [9] Nguyen, A. Tuan, Philip Torr, and Ser Nam Lim. "Fedsr: A simple and effective domain generalization method for federated learning." Advances in Neural Information Processing Systems 35 (2022): 38831-38843.
>
> [10] Chen, Haokun, et al. "Fraug: Tackling federated learning with non-iid features via representation augmentation." Proceedings of the IEEE/CVF International Conference on Computer Vision. 2023.
>
> ---
>
> **Q:** Doesn't that mean a related generative model has to exist for a given problem? What if no generative model exists for a given problem?
>
> **Response:**
>
> Currently, there has been many generative models for many domains, which is why we think it is interesting and timely to explore this topic.
>
> Besides, the generative model does not neccessarily need to fully resemble real data. Here are our evidence.
>
> 1. The experiments on **medical dataset HAM10000 in Table 4**. Since stable diffusion can not generate medical with good quality, we choose this experiment to alleviate the concern of contamination. Please note that the generated images are completely different from the real iamges and we decided not to posting them to avoid causing discomfort. Here, we put the results below for convinience. **Despite the dissimilarity, our method still brings significant gain.**
>
> [**Table R2.** Results on medical dataset.]
> | Baseline | Without FedGC | With FedGC |
> | :------: | :-----------: | :--------: |
> | FedAvg   | 48.57         | **56.67**      |
> | FedProx  | 49.52         | **56.19**      |
> | SCAFFOLD | 54.76         | **58.57**      |
>
> 2. The experiments of training on **generated data only in Table 6.** Here, we put the results below for convinience. From the table, we see that **merely using generative data to train a model fail to perform well**, indicating that there is a huge gap between generative data and real data. Generated data can only exhibit its effectiveness when used in conjunction with real data in our proposed FedGC.
>
> [**Table R3.** Comparisons with training on genrative data only. Training on generative data only achieves low performance.]
> | Baseline | Real Data | Generative Data | FedGC (ours) |
> |:--------:|:---------:|:---------------:|:------------:|
> |  FedAvg  |   60.77   |      41.85      |  **73.99**   |
> | FedProx  |   63.62   |      40.93      |  **73.69**   |
> | SCAFFOLD |   65.00   |      43.45      |  **75.79**   |
>
> 3. We compare real data and generative data in Figure 8 via visualization, where we see that there is a large amount of generative data with incorrect concept and can not fully represent real data.
>
>
> ---
>
> Overall, we thank the reviewer for the time for reviewing. We hope that our responses can fully address your concerns. We also kindly request the reviewer for reconsideration and look forward to your feedback to improve our work.

---

> ### Comment · Reviewer_kS7r · 2023-11-22
> **Response to W2**
>
> I thank the authors for their comments. They largely address my concerns. Unfortunately due to time constraints I will not be able to discuss all points at once. I will try to tackle one by one.
>
> Regarding your response to W2, maybe I was not clear in explaining my concern. The problem of data heterogeneity is very unique to FL. This is a major problem. I am aware of that. But it is not what I was concerned. I was concerned with the proposed solution where the data generation process is almost independent from one client to another. I was hoping that FedGC would do something smarter than existing FL approaches such that it can somehow pass some knowledge related to its local data distribution or the knowledge of how the local generated data were generated to other clients in the hope that they can use that knowledge to learn their own classification problem better. But it would not. Hence, it appears the federated setting here is unnecessary to the idea that the paper is promoting, which is to use generative models to generated data, other than to bring the data heterogeneity problem into the context.
>
> In the new Table R1 you provided, do you have a row representing the results of local training without using generated content? How do they compare to Local+GC results? I suspect Local+GC should outperform Local (without GC). Going from Local to Local+GC can tell us how much the gain comes from generated content alone, and then from Local+GC to FedGC can tell us further how much we get when the gradients are communicated.

---

> > ### Author Response · Authors · 2023-11-22
> > **Thanks for the feedback.**
> >
> > Thank you for your feedback and time. Here are our detailed responses.
> >
> > ---
> >
> > **Concern 1-1:** Data generation process is almost independent from one client to another, it can somehow pass some knowledge related to its local data distribution.
> >
> > **Response:**
> >
> > We truly thank the reviewer for the advice. However, we acturally have ever thought about this but decided NOT to do so for the following reasons.
> >
> > 1. Passing knowledge related to local data distribution could introduce **additional communication cost and privacy concern** since the local data distribution itself can be sensitive information. However, as pointed by one of the most influential survey paper [1], it is exactly stated that **"privacy and communication efficiency are always first-order concerns in FL"** (at the second paragraph of page 10 in https://arxiv.org/pdf/1912.04977.pdf). Since most of the authors of this paper are quite representative in FL community, we choose to follow their advice.
> >
> > 2. Over the last few years, we have seen so many samples where the authors are punished because their methods require communicating additional information (for privacy and communication issue). So, that is the lesson we have learnt, guiding us to design a method without transmitting any information. We really need your understanding. Otherwise, we do not know what should we do to meet the standard of reviewers. To transmit additional information (be questioned for privacy and communication issue) or not (be questioned for simplicity)?
> >
> > [1] Kairouz, Peter, et al. "Advances and open problems in federated learning." Foundations and Trends® in Machine Learning 14.1–2 (2021): 1-210.
> >
> > ---
> >
> > **Concern1-2:** It appears the federated setting here is unnecessary to the idea that the paper is promoting, which is to use generative models to generated data, other than to bring the data heterogeneity problem into the context.
> >
> > **Response:**
> >
> > We would like to emphasize that our method of using generative content is organically suited for data heterogeneity issue in federated learning. We kindly request to reviewer to refer to our **Figure 11 for a strong evidence**. From the figure, we can see that:
> >
> > 1. When the data heterogeneity level is low, baseline with generative content only achieves comparable performance as baseline.
> >
> > 2. When the data heterogeneity level is high, baseline with generative content achieves significantly better than baseline.
> >
> > These two observations provide strong evidence that generative content can really play a key role in the heterogeneous setting in FL. While for non-heterogeneous FL or non-FL setting where there is minor or even no data heterogeneity, it appears that generative content cannot bring much benefit.
> >
> > Thus, in our paper, **generative content is indeed effective for and highly-related to data heterogeneity issue in FL**.
> >
> > ---
> >
> > **Concern 2:** Do you have a row representing the results of local training without using generated content?
> >
> > **Response:**
> >
> > There is some diffuculty due to limited time and resources. However, we will try our best to show the results as soon as possible.
> >
> > However, we can already see clear incentive for a client to participate federated learning especially using generative content. This is because that from Table R1, we can see that FedGC can significantly outperforms Local+GC. For example, under EuroSAT-High, FedGC even achieves 50% better than Local+GC! And Local+GC only has a performance of 24.87%, which is almost useless! This shows that even if GC brings gain to Local, Local+GC still can not work well, providing strong motivation for FedGC.

---

> > > ### Author Response · Authors · 2023-11-22
> > >
> > > **Concern 3:** The problem I see is that it only says that for these datasets FedGC works. Without some sort of theoretical statements about the generalisation of FedGC to other datasets, it is hard to understand how well FedGC would work in a new setting.
> > >
> > > **Response:**
> > >
> > > We want to emphasize that we have considered **seven** datasets, including CIFAR10, EuroSAT, PACS, VLCS, HAM10000, Sentiment140, and Yahoo! Answers. They have covered multiple heterogeneity types, heterogeneity levels, domains, modalities. As a reference, previous papers **often consider 2-5 datasets [1,2,3,4,5,6,7,8,9,10]**.
> > >
> > > As the first work to consider mixing private data with generative content in federated learning. We focus on providing a systemetic empirical study throughout the paper, as evidented by diverse datasets, settings, and ablations in paper (including appendix). Given that we have made huge efforts on experiments, it is hard to consider both empirical and theoretical results in a 9-page paper. We will continue to explore this topic including theory in the future as we have found several interesting findings that the current theory cannot explain (please refer to Figure 6 and its discussion; also Section 5).
> > >
> > > [1] Li, Qinbin, Bingsheng He, and Dawn Song. "Model-contrastive federated learning." Proceedings of the IEEE/CVF conference on computer vision and pattern recognition. 2021.
> > >
> > > [2] Chen, Hong-You, et al. "On the importance and applicability of pre-training for federated learning." The Eleventh International Conference on Learning Representations. 2023.
> > >
> > > [3] Nguyen, John, et al. "Where to Begin? On the Impact of Pre-Training and Initialization in Federated Learning." The Eleventh International Conference on Learning Representations. 2023.
> > >
> > > [4] Oh, Jaehoon, SangMook Kim, and Se-Young Yun. "FedBABU: Toward Enhanced Representation for Federated Image Classification." International Conference on Learning Representations. 2022.
> > >
> > > [5] Thapa, Chandra, et al. "Splitfed: When federated learning meets split learning." Proceedings of the AAAI Conference on Artificial Intelligence. Vol. 36. No. 8. 2022.
> > >
> > > [6] Yang, Fu-En, Chien-Yi Wang, and Yu-Chiang Frank Wang. "Efficient model personalization in federated learning via client-specific prompt generation." Proceedings of the IEEE/CVF International Conference on Computer Vision. 2023.
> > >
> > > [7] Pennisi, Matteo, et al. "Experience Replay as an Effective Strategy for Optimizing Decentralized Federated Learning." Proceedings of the IEEE/CVF International Conference on Computer Vision. 2023.
> > >
> > > [8] Zhang, Ruipeng, et al. "Federated domain generalization with generalization adjustment." Proceedings of the IEEE/CVF Conference on Computer Vision and Pattern Recognition. 2023.
> > >
> > > [9] Nguyen, A. Tuan, Philip Torr, and Ser Nam Lim. "Fedsr: A simple and effective domain generalization method for federated learning." Advances in Neural Information Processing Systems 35 (2022): 38831-38843.
> > >
> > > [10] Chen, Haokun, et al. "Fraug: Tackling federated learning with non-iid features via representation augmentation." Proceedings of the IEEE/CVF International Conference on Computer Vision. 2023.
> > >
> > > ---
> > >
> > > Thank you for your tendency to raise your rating. We hope that our responses can fully address your concerns and look forward to your feedback to improve our work.

---

> > > > ### Comment · Reviewer_kS7r · 2023-11-22
> > > > **Thank you. I've updated my ratings.**
> > > >
> > > > Thank you for your very prompt but detailed comments. All my concerns are properly addressed.
> > > >
> > > > I would like to thank the authors for having been patient with me throughout the rebuttal period and politely corrected me where I was wrong. All your comments are very helpful. I clearly overlooked this good paper during the review period. No excuse for that. I have updated my overall rating to 8. Good paper.

---

> > > > > ### Author Response · Authors · 2023-11-22
> > > > > **Tremendous thanks!**
> > > > >
> > > > > We really appreciate your understanding and recognition, which truly gave us tremendous courage to continue exploring this topic. Thanks for your time and useful feedback!

---

> ### Comment · Reviewer_kS7r · 2023-11-22
> **Response to W1/W3**
>
> W1: I am glad that you acknowledge that federated learning itself is a practical setting. While you have shown that in your datasets, the experimental results when additional generated content was used increase the accuracy, the problem I see is that it only says that for these datasets FedGC works. Without some sort of theoretical statements about the generalisation of FedGC to other datasets, it is hard to understand how well FedGC would work in a new setting.
>
> W3: You are right. I have overlooked. I will increase my rating.

---

### Official Review · Reviewer_c8uL · 2023-10-28

**Soundness:** 2 fair
**Presentation:** 3 good
**Contribution:** 2 fair
**Rating:** 6
**Confidence:** 3

**Summary:**

The paper explores whether the data heterogeneity issues in federated learning (that limits its performance) could be mitigated by adding synthetic data generated via generative models. The authors first conduct experiments on two image datasets (CIFAR-10 and EuroSAT) and two language datasets (PACS and VLCS) to show that the performance of federated learning significantly improves after combining the private training data with new data that are generated based on the guidance of prompts and private data simultaneously. Interestingly, the authors also observe reduced privacy risk in FL after adding generated data, where the privacy risk means the average success of simple loss-based membership inference attacks over different clients. Algorithmically, the authors conduct extensive experiments and attributed the success of the method to four critical choices: amount of generative contents, the same amount of generative contents for each user, using multiple prompts to guide data generation; and simultaneously use text prompts and real private data for generation.

To further understand the reason for the performance gain, the authors additionally conduct several interesting ablation studies, where the central conclusions are:
- The performance improvement exists even when the generated data is not similar to the private data.
- Adding generative contents reduced data heterogeneity, and under a higher amount of generative contents, the performance of FedAvg becomes better or on par with other algorithms that are designed to tackle data heterogeneity (such as FedProx and SCAFFOLD).
- Adding generative contents reduces the client drift effect in FL.

**Strengths:**

- Thorough experiments that investigate the algorithmic choices and how federated learning is affected under new generative data.
- Mitigating the issues due to data heterogeneity in FL is an important question, and exploring how large generative models could alleviate such issues is a timely and vital direction.

**Weaknesses:**

- On the one hand, the authors show that increasing the amount of generative data always increases the learning performance (Table 2 and Figure 2). On the other hand, the authors also show that FL on only generative contents (no private data) performs poorly in Table 5. This seems counterintuitive. Could the authors explain why?

- If we allow each user to train a local model on its private data combined with generative contents, would the performance be comparable to FL training on private data combined with newly generated data? If so, there would not be any incentive for clients to perform FL when they have access to additional generative content, thus deeming the problem setting as insignificant.

**Questions:**

- See weakness for two questions.

- Additionally, could authors discuss the possible dataset contamination? That is, whether the benchmark dataset is already used in training dataset for the generative model.

---

> ### Author Response · Authors · 2023-11-21
>
> Thank you for your time for reviewing and your comments. Here are our detailed responses.
>
> ---
>
> **W1:** On the one hand, the authors show that increasing the amount of generative data always increases the learning performance. On the other hand, the authors also show that FL on only generative contents performs poorly in Table 5. This seems counterintuitive. Could the authors explain why?
>
> **Response:**
>
> That is a good point. Acturally, **they are not counterintuitive. Please note that in Table 2, the performance of using 50000 samples is worse than performance of using 20000 samples.**
>
> Here is the rationale. (1) First, the results showing that only generative contents performs poorly indicates that the generative data cannot fully represent real data. (2) Second, when generative data is combined with real heterogeneous data, the generative data serves to mitigate data heterogeneity and overfitting issue. Therefore, the performance is increased after adding generative data.
>
> However, when the number of generative samples is too large, the issue of dissimilarity between generative and real data tends to bring negative effects.
>
>
> [**Table R1.** Performance under different number of generative samples.]
> | Baseline |   0   | 1000  | 10000 |   20000   | 50000 |
> |:--------:|:-----:|:-----:|:-----:|:---------:|:-----:|
> |  FedAvg  | 61.25 | 66.98 | 74.50 | **76.93** | 76.39 |
> | FedProx  | 64.02 | 68.55 | 74.36 | **76.81** | 76.73 |
> | SCAFFOLD | 63.98 | 71.33 | 73.96 | **74.88** | 73.98 |
>
> ---
>
> **W2:** If we allow each user to train a local model on its private data combined with generative contents, would the performance be comparable to FL training on private data combined with newly generated data?
>
> **Response:**
>
> Thanks for this insightful comment. Following your advice, we have conducted the following experiments. From the table, we see that there is still a **huge gap between local training and FedGC**, thus incentivizing clients to participate federated learning.
>
> [**Table R2.** Performance comparison between local training with generative content and our FedGC.]
> | Method | CIFAR-High | CIFAR-Low | EuroSAT-High | EuroSAT-Low |
> |:--------:|:----------:|:---------:|:------------:|:-----------:|
> | Local+GC |   46.89   |   50.47   |    24.87     |    35.48    |
> |  FedGC   | **74.50**  | **79.93** |  **74.83**   |  **84.46**  |
>
> ---
>
> **Q:** Additionally, could authors discuss the possible dataset contamination? That is, whether the benchmark dataset is already used in training dataset for the generative model.
>
> **Response:** Sure, we also take this factor seriously and that is why we have shown the following results in the submission.
>
>
> 1. The experiments on **medical dataset HAM10000 in Table 4**. Since stable diffusion can not generate medical with good quality, we choose this experiment to alleviate the concern of contamination. Please note that the generated images are completely different from the real iamges and we decided not to posting them to avoid causing discomfort. Here, we put the results below for convinience. **Despite the dissimilarity, our method still brings significant gain.**
>
> [**Table R1.** Results on medical dataset.]
> | Baseline | Without FedGC | With FedGC |
> | :------: | :-----------: | :--------: |
> | FedAvg   | 48.57         | **56.67**      |
> | FedProx  | 49.52         | **56.19**      |
> | SCAFFOLD | 54.76         | **58.57**      |
>
> 2. The experiments of training on **generated data only in Table 6.** Here, we put the results below for convinience. From the table, we see that **merely using generative data to train a model fail to perform well**, indicating that there is a huge gap between generative data and real data. Generated data can only exhibit its effectiveness when used in conjunction with real data in our proposed FedGC.
>
> [**Table R2.** Comparisons with training on genrative data only. Training on generative data only achieves low performance.]
> | Baseline | Real Data | Generative Data | FedGC (ours) |
> |:--------:|:---------:|:---------------:|:------------:|
> |  FedAvg  |   60.77   |      41.85      |  **73.99**   |
> | FedProx  |   63.62   |      40.93      |  **73.69**   |
> | SCAFFOLD |   65.00   |      43.45      |  **75.79**   |
>
> 3. We compare real data and generative data in Figure 8 via visualization, where we see that there is a large amount of generative data with incorrect concept and can not fully represent real data.
>
> ---
>
> Overall, we hope that our responses can fully address your concerns and will be grateful for any feedback.

---

> ### Author Response · Authors · 2023-11-22
> **We sincerely anticipate your feedback as the Discussion phase will conclude in 16 hours.**
>
> Dear Reviewer,
>
> We have carefully considered your comments and provided further clarifications and explanations in detail. We have:
>
> - Shown clear evidence that the results in Table 2 and Table 6 are not counterintuitive.
> - Provided experimental results to verify sufficient incentive for clients to participate FL.
> - Provided our detailed discussion on dataset contamination.
>
>  As Discussion phase will end in 16 hours, we would be grateful if you could check our responses and reconsider your rating.
>
> Best regards,
>
> Authors

---

> > ### Comment · Reviewer_c8uL · 2023-11-23
> >
> > Thanks for the response. Most of my concerns are addressed. I've increased the score. Minor comments about the comparison between local training + generative data versus FL + generative data, it would be good to know whether the total number of generative data is kept the same for each client.

---

> > > ### Author Response · Authors · 2023-11-23
> > > **Thanks for your feedback and recognition!**
> > >
> > > We really appreciate your feedback and recognition! Your valuable comments indeed help us improve our work.
> > >
> > > For your minor comments, the answer is: yes the total number of generative data is kept the same for each client, making them a fair comparison.

---

### Official Review · Reviewer_kHC9 · 2023-10-31

**Soundness:** 3 good
**Presentation:** 3 good
**Contribution:** 2 fair
**Rating:** 3
**Confidence:** 5

**Summary:**

This paper proposed FedGC, a synthetic-data-based federated learning system. In high-level speaking, FedGC utilized the foundational model on the local side to generate synthetic data, and mixed the synthetic data with real private data for local training. The experiments with both language and computer vision benchmark datasets show the effectiveness of FedGC.

**Strengths:**

1. It is interesting to utilize the power of foundational models to assist federated learning.

2. The experiment and ablation study are detailed.

**Weaknesses:**

1. The author does not consider the generation cost in the paper. The Stable Diffusion model needs at least 4.2GB space to deploy locally, and the memory consumption of generation is huge for the IoT or cross-device FL setup. For the black-box foundational models such as ChatGPT, the prompts directly leak the data privacy to the server of ChatGPT. As a result, both methods do not fit the FL setups.

2. The mixed-up training of synthetic and real private data directly increases the computational burden for the local devices.

3. I am concerned about the in-domain generation problem. As the Stable Diffusion is trained with the LAION-5B dataset, we cannot guarantee that the Stable Diffusion does not meet with the test data, such as the CIFAR dataset, during the training. As a result, we could not distinguish whether the performance boost-up is coming from the FedGC or the in-domain generation of the foundational models.

**Questions:**

1. How many synthetic data samples does the local client generate for the experiment in Table 1?

2. During local data generation, does the local client only generate the data for the label it holds or generate the data for the whole label space among all participants?

---

> ### Author Response · Authors · 2023-11-21
>
> Thank you for your time and comments. Here are our responses in detail.
>
> ---
>
> **W1:** The author does not consider the generation cost in the paper. The Stable Diffusion model needs at least 4.2GB space to deploy locally, and the memory consumption of generation is huge for the IoT or cross-device FL setup. For the black-box foundational models such as ChatGPT, the prompts directly leak the data privacy to the server of ChatGPT. The method does not fit the FL setups?
>
> **Response:**
>
> Foremost, we want to reiterate the emphasis of our paper: (1) the **idea** of leveraging advanced generative models to assist training on private data, and (2) the **general framework** with diverse solutions for each sub-step.
>
> Regarding the concern of generation cost, it can be easily alleviated by generating data using **online interfaces (rather than generating locally)**, which are commonly available right now (e.g., huggingface) [1]. And we can generate generate data without real-data guidance for such cases since our proposed FedGC provides diverse solutions for each generation step. In this case, **both the issues of generation burden and privacy issue are alleviated.**
>
> Besides, we acturally have recognized such potential concern so that we **do not use real-data-guidance for all the experiments except Table 4**. However, since our focus is to propose a general and multifunctional framework, we still decided to include this because such strategy can bring additional benefit if the computational resource is available.
>
> [1] https://huggingface.co/runwayml/stable-diffusion-v1-5
>
> ---
>
> **W2:** The mixed-up training of synthetic and real private data directly increases the computational burden for the local devices?
>
> **Response:** There could be some misunderstanding. **Mixed training does not increase the computational burden.**
>
> Please note that throughout our experiments, for each round, we are running with **the sample number of SGD iterations** for all clients and methods. That is, after mixing data, the used batch size and the number of SGD iterations are exactly the sample as baselines.
>
> ---
>
> **W3:** I am concerned about the in-domain generation problem. We could not distinguish whether the performance boost-up is coming from the FedGC or the in-domain generation of the foundational models.
>
> **Response:** Acturally, we have put efforts to alleviate such concerns from three aspects.
>
> 1. The experiments on **medical dataset HAM10000 in Table 4**. Since stable diffusion can not generate medical with good quality, we choose this experiment to alleviate the concern of in-domain generation. Please note that the generated images are completely different from the real iamges and we decided not to posting them to avoid causing discomfort. Here, we put the results below for convinience. **Despite the dissimilarity, our method still brings significant gain.**
>
> [**Table R1.** Results on medical dataset.]
> | Baseline | Without FedGC | With FedGC |
> | :------: | :------: | :------: |
> | FedAvg   | 48.57         | **56.67**      |
> | FedProx  | 49.52         | **56.19**      |
> | SCAFFOLD | 54.76         | **58.57**      |
>
> 2. The experiments of training on **generated data only in Table 6.** Here, we put the results below for convinience. From the table, we see that **merely using generative data to train a model fail to perform well**, indicating that there is a huge gap between generative data and real data. Generated data can only exhibit its effectiveness when used in conjunction with real data in our proposed FedGC.
>
> [**Table R2.** Comparisons with training on genrative data only. Training on generative data only achieves low performance.]
> | Baseline | Real Data | Generative Data | FedGC (ours) |
> |:---:|:---:|:---:|:---:|
> |  FedAvg  |   60.77   |      41.85      |  **73.99**   |
> | FedProx  |   63.62   |      40.93      |  **73.69**   |
> | SCAFFOLD |   65.00   |      43.45      |  **75.79**   |
>
> 3. We compare real data and generative data in Figure 8 via visualization, where we see that there is a large amount of generative data with incorrect concept and can not fully represent real data.
>
> ---
>
> **Q1:** How many synthetic data samples does the local client generate for the experiment in Table 1?
>
> **Response:** The total amount for all clients is 20% of the original dataset. For example, CIFAR-10 has 50000 samples and then we will generate 10000 samples in total. Since there is 10 clients, each client will have 1000 samples.
>
> ---
>
> **Q2:** During local data generation, does the local client only generate the data for the label it holds or generate the data for the whole label space among all participants?
>
> **Response:** Local client generates data for the whole label space.
>
> ---
>
> Overall, we are so regret to see that the reviewer gave such a rating for our new exploration in federated learning. We hope that our responses can fully address the reviewer's concerns and would be grateful to get any useful feedback to improve our work.

---

> ### Author Response · Authors · 2023-11-22
> **We sincerely anticipate your feedback as the Discussion phase will conclude in 16 hours.**
>
> Dear Reviewer,
>
> We have carefully considered your comments and provided further clarifications and explanations in detail. We hope that we have well address them through the provided methodological and empirical evidence. As Discussion phase will end in 16 hours, we would be grateful if you could check our responses and reconsider your rating.
>
> Best regards,
>
> Authors

---

> > ### Comment · Reviewer_kHC9 · 2023-11-22
> >
> > Thank you for your response. I still have concerns about the paper.
> >
> > W1: If the generation could be done on the cloud side, the communication cost would be very high for the client for transmitting the synthetic data. As a result, it is still unfeasible for cross-device FL setups due to the large communication burden.
> >
> > Q2: How would the local client know the whole label space? To be specific, under the non-iid distribution, each client has a skewed label distribution for the local data. How would the local client know the labels that it does not hold locally? If the client wants to hold whole label space information, some client-wise communication is needed and some privacy issues may be raised as well.
> >
> > As a result, I prefer not to change my score.

---

> > > ### Author Response · Authors · 2023-11-23
> > >
> > > **Q1:** How would the local client know the whole label space? To be specific, under the non-iid distribution, each client has a skewed label distribution for the local data. How would the local client know the labels that it does not hold locally?  If the client wants to hold whole label space information, some client-wise communication is needed and some privacy issues may be raised as well.
> > >
> > > **Response:**
> > >
> > > In federated learning, clients should share the same label space even if the client has a skewed label distribution. The server should send the task description which includes the whole label space to all clients. **This is a basic requirement for federated learning, rather than an additional requirement introduced by our method.** Here are the reasons.
> > >
> > > 1. For tasks such as image classification, the server should send the mapping relationship between object and label (e.g., airplane corresponds to label 0, automibile corresponds to label 1) to all clients. Only with this mapping relationship, each client can process its data for this task. Otherwise, there will be two key issues.
> > >     - **Inconsistent label space.** For example, both two clients have the airplane category, however, client A assigns airplane as label 0 while client B assigns airplane as label 1. This can significantly harm performance of the model.
> > >     - **Introducing unrelated data.** One client could have data of diverse categories, where some of them may be unrelated to the current FL task. Suppose the current FL task is a 10-way classification task while client A has 2 categories within these 10 categories but also have 2 unrelated categories out of these 10 categories. In this case, if the server does not send the whole label space to all clients, then how could the client assign label for the data?
> > >
> > > 2. For tasks such as next word prediction, the server should send the token-to-label or word-to-label mapping relationship to all clients. For example, all clients should be consistent on that the word 'hello' corresponds to label 0 and 'world' corresponds to label 1. Otherwise, there is no way to train a next word prediction model.
> > >
> > > **Therefore, sharing the whole label space is the fundamental requirement for federated learning and our method does not introduce any privacy issue.**
> > >
> > > ---
> > >
> > > We hope that our responses can fully address your concerns and look forward to your feedback.

---

> > > ### Author Response · Authors · 2023-11-23
> > > **We sincerely anticipate your feedback as the Discussion phase will conclude in half an hour.**
> > >
> > > Dear Reviewer,
> > >
> > > We have carefully considered your comments and provided further clarifications and explanations in detail. Sorry for the potential confusion previously and we believe that we have well addressed your concerns, including showing that the **our method can significantly outperform baseline with less communication cost!**
> > >
> > >  We would be grateful if you could check our responses and reconsider your rating.
> > >
> > > Best regards,
> > >
> > > Authors

---

> ### Author Response · Authors · 2023-11-23
> **Thanks for the feedback.**
>
> Thanks for the further comments and here are our responses.
>
> ---
>
> **W1:** If the generation could be done on the cloud side, the communication cost would be very high for the client?
>
> **Response:** Even in such cases, the communication cost is **not HIGH** as stated by the reviewer. On the contrary, the communication cost is **quite LOW**.
>
> 1. Even in such cases, the additional communication cost is acturally minor and the the enhanced performance is significant. Here, we provide a detailed example on launching FedGC on SCAFFOLD on CIFAR10 in the following table. From the table, we can see that FedGC can achieve significantly higher performance than the baseline while introducing minor additional communication cost. Besides, please note that we only introduce some downlink cost rather than uplink cost, and it is commonly known that the uplink is slower at least five times than the downlink [1,2]. Specifically, **FedGC can achieve 5.07% absolute accuracy improvement while only introducing 0.007% additional communication cost!**
>
> &emsp;
>
> [**Table R3.** Communication cost per client and accuracy in cases where we use cloud generation.]
> |      Method       |  SCAFFOLD   | FedGC-100 | FedGC-200 | FedGC-1000 | FedGC-10000 |
> |:-----------------:|:-----------:|:---------:|:---------:|:----------:|:-----------:|
> | Downlink Cost (B) | 215,777,600 |  +30,720  |  +61,400  |  +307,200  | +3,072,000  |
> |  Uplink Cost (B)  | 215,777,600 |    +0     |    +0     |     +0     |     +0      |
> |  Total Cost (B)  | 431,555,200 |  +30,720  |  +61,400  |  +307,200  | +3,072,000  |
> |  Additional Cost (%)  | - |  +0.007%  |  +0.014%  |  +0.071%  | +0.712%  |
> |     Accuracy      |    63.98    |  **+5.07%**   |  **+7.35%**   |   **+7.35%**   |   **+9.98%**    |
>
> &emsp;
>
> 2. To further alleviate the reviewer's concern, we provide the following table where we keep the communication cost less than baselines by reducing the communication rounds (i.e., 1-2 rounds reduction) for FedGC. From the table, **we see that even with less communication cost, FedGC still significantly outperforms the baseline!**
>
> &emsp;
>
> [**Table R4.** Accuracy comparison between FedGC and SCAFFOLD when keeping FedGC with less communication cost.]
> | Method   | SCAFFOLD | FedGC-100 | FedGC-200 | FedGC-1000 | FedGC-10000 |
> | :------: | :------: | :----: | :-------: | :---: | :----: |
> |  Total Cost (B)  | 431,555,200 |  427,270,368  |  427,301,048  |  427,546,848  | 430,311,648  |
> | Accuracy | 63.98    | **69.05%**    | **71.33%**    | **71.33%**     | **73.96%**            |
>
> &emsp;
>
> [1] Yi, Liping, Wang Gang, and Liu Xiaoguang. "QSFL: A two-level uplink communication optimization framework for federated learning." ICML 2022.
>
> [2] Konečný, Jakub, et al. "Federated learning: Strategies for improving communication efficiency." arXiv preprint arXiv:1610.05492 (2016).

---

### Meta-Review · Area_Chair_EmuW · 2023-12-10

**Metareview:**

This paper introduces FedGC, a synthetic-data-based federated learning system aimed at addressing data heterogeneity issues. In FedGC, foundational models on local devices generate synthetic data, which is then mixed with real private data for local training. The authors then investigate experimentally whether the addition of synthetic data can mitigate data heterogeneity issues in federated learning. Experiments on image datasets (CIFAR-10 and EuroSAT) and language datasets (PACS and VLCS) demonstrate enhanced federated learning performance when combining private training data with generated data. Notably, the authors observe a reduced privacy risk in federated learning after the inclusion of generated data, attributing the success of the method to four critical choices: the amount of generative content, uniform distribution of generative contents for each user, the use of multiple prompts for data generation guidance, and simultaneous use of text prompts and real private data for generation.

This novelty of the paper hinges on the practicality of the proposed solution using generative content for local devices in federated learning, as the primary contributions are dominantly engineering, which is fine but also raises the bar for the evaluation. The paper can benefit from another round of revision to fairly show the benefits of the proposed solution. Using generative content on local devices sounds like a cool idea, but inevitably introduces computation (running a generative model), storage, and communication (potential data communication between devices and online hosted models) challenges for devices. On the other hand, if a local client is able to defeat all the above-listed challenges, it seems we will then be operating in a situation where federal learning won’t be entirely necessary. One reviewer also pointed out practical challenges in obtaining additional data distribution information in order to fully leverage the generative model.

**Justification For Why Not Higher Score:**

While the introduced concept and demonstration are promising, the paper does need a stronger justification and experiments to fairly show the benefits of the proposed solution.

**Justification For Why Not Lower Score:**

N/A

---

### Decision · Program_Chairs · 2024-01-16

Reject